# A Neural Pre-Conditioning Active Learning Algorithm to Reduce Label Complexity

**Seo Taek Kong**[1,*,†]    **Soomin Jeon**[2]    **Dongbin Na**[3]    **Jaewon Lee**[3]    **Hong-Seok Lee**[3]
**Kyu-Hwan Jung**[4,*,†]

[1]University of Illinois, Urbana-Champaign    [2]Dong-A University    [3]VUNO Inc.
[4]Sungkyunkwan University

## Abstract

Deep learning (DL) algorithms rely on massive amounts of labeled data. Semi-supervised learning (SSL) and active learning (AL) aim to reduce this label complexity by leveraging unlabeled data or carefully acquiring labels, respectively. In this work, we primarily focus on designing an AL algorithm but first argue for a change in how AL algorithms should be evaluated. Although unlabeled data is readily available in pool-based AL, AL algorithms are usually evaluated by measuring the increase in supervised learning (SL) performance at consecutive acquisition steps. Because this measures performance gains from both newly acquired instances and newly acquired labels, we propose to instead evaluate the label efficiency of AL algorithms by measuring the increase in SSL performance at consecutive acquisition steps. After surveying tools that can be used to this end, we propose our neural pre-conditioning (NPC) algorithm inspired by a Neural Tangent Kernel (NTK) analysis. Our algorithm incorporates the classifier's uncertainty on unlabeled data and penalizes redundant samples within candidate batches to efficiently acquire a diverse set of informative labels. Furthermore, we prove that NPC improves downstream training in the large-width regime in a manner previously observed to correlate with generalization. Comparisons with other AL algorithms show that a state-of-the-art SSL algorithm coupled with NPC can achieve high performance using very few labeled data.

## 1 Introduction

Active learning (AL) describes the setting where a model can interact with a dedicated annotator and query for labels. This is in contrast to passive learning where labels are acquired randomly. In pool-based AL, both unlabeled $\mathcal{Z}_U = \mathcal{X}_U$ and labeled data $\mathcal{Z}_L = (\mathcal{X}_L, \mathcal{Y}_L)$ are available for training, and labels are incrementally acquired by querying $|\mathcal{Y}^s| = Q$ labels at each query step $s$ until a labeling budget $B$ is met. The question we aim to answer in this work is: "Given images $\mathcal{X}$ and a labeling budget $B$, what's the maximum performance that can be achieved?" Traditionally, AL algorithms have been evaluated by measuring downstream supervised learning (SL) performance, i.e. training on $\mathcal{Z}_L$, but we argue that downstream semi-supervised learning (SSL) performance, i.e. training on $\mathcal{Z} = \mathcal{Z}_L \cup \mathcal{Z}_U$, is a better benchmark for the following reasons. Supervised learning performance as an evaluation metric for AL fails to extricate performance gains from newly-acquired labels $\mathcal{Y}^{s+1}$ from the influence of newly-acquired samples $\mathcal{X}^{s+1}$. In contrast, SSL trains a model on $\mathcal{Z}$ and the only difference of datasets $\mathcal{Z}^{s+1} - \mathcal{Z}^s$ after subsequent query steps is the labels $\mathcal{Y}^{s+1}$, and the respective performances reveal gains from newly-acquired labels. While it has been shown

---

\* This work was submitted while the authors worked at Vuno Inc.

† Correspondence to skong10@illinois.edu and khwanjung@skku.edu

that many AL algorithms outperform passive learning (PL) with respect to instance-label efficiency, we see that some AL algorithms in fact under-perform PL with respect to this criterion that measures label efficiency.

Simply replacing SL with SSL can introduce new problems in evaluating the label-efficiency of AL algorithms. While in principle more data should always be better, SSL performance deteriorates significantly when the labeled set's classes are imbalanced (Lee et al., 2021; Kim et al., 2020). In AL, the number of images corresponding to each class cannot be observed prior to labeling, and constructing a balanced labeled set would require discarding majority classes retrospectively. Class imbalance had not been as problematic when benchmarking AL algorithms with SL performance because large query sizes ultimately yields class distributions closer to uniform. We address this issue by adopting the recently-proposed distribution re-alignment method (Kim et al., 2020) applied to a widely used SSL algorithm called FixMatch (Sohn et al., 2020).

Having motivated downstream SSL performance as an evaluation metric for AL, we seek to maximize performance with respect to a labeling budget $B$. We propose a neural pre-conditioning (NPC) algorithm that builds on prior work and addresses problems noticed in literature. Our algorithm uses the Gram matrix of the model's gradients with respect to parameters. Gradients of the loss function have been used by Huang et al. (2016); Ash et al. (2020) in the context of AL to model a classifier's uncertainty about unlabeled samples, whereas we use gradients of the classifier's outputs. Because of the difference, the embeddings used by NPC capture uncertainty by comparing the direction of gradients with other possibly more certain samples' in addition to their magnitudes. Moreover, our algorithm operates in the batch-mode setting where the collective importance of candidate samples is measured together. Lastly, we show that the landscape of downstream SSL is improved when supervised on labeled data selected by NPC which is why we name it neural pre-conditioning.

This paper is structured as follows. Section 2 formally describes why comparing downstream SL performance after consecutive acquisition steps fails to measure the label-efficiency of AL algorithms, and proposes to consider downstream SSL performance as a fair evaluation metric. Section 3 lays out the observations made in prior works that motivate the proposed algorithm before proceeding to stating the algorithm and how it addresses these concerns. Section 4 addresses the last pre-conditioning property of the algorithm and describes potential benefits to a randomized search procedure invoked by NPC. Lastly, Section 5 presents experiments that show how the proposed algorithm enhances downstream SSL performance, and highlights how some AL algorithms are not as effective in our proposed setting.

## 2 Problem Setting and Related Work

### 2.1 Active Semi-Supervised Learning

To measure performance gains from only newly acquired labels, we alternate between applying AL to acquire labels and training a classifier on newly acquired data using SSL, where at first a small set of labels $|\mathcal{Y}_L^0| = Q_0$ with balanced classes is assumed. At each query step $s$, the classifier queries for $Q$ labels $\mathcal{Y}^s$ corresponding to samples $\mathcal{X}^s$ from the remaining unlabeled pool $\mathcal{X}_U$. A state-of-the-art SSL algorithm named FixMatch (Sohn et al., 2020) with a pseudo-label refinement procedure (DARP, Kim et al. (2020)) is used to handle class imbalance when training after subsequent query/acquisition steps. We refer to the above procedure as active semi-supervised learning (ASSL) following (Hanneke, 2007), and remark that the term has been used to refer to different procedures (Wang et al., 2016). Our ASSL setting closely follows standard AL benchmarks (Ash et al., 2020; Sener & Savarese, 2017; Gissin & Shalev-Shwartz, 2019) with the difference being that SSL, instead of SL, is invoked for training.

We explain why comparing downstream SSL, instead of SL, performance at consecutive acquisition steps is a better evaluation scheme when measuring the label-efficiency of AL algorithms. Consider two fully-trained classifiers at consecutive acquisition steps $s$ and $s + 1$. The performance difference of downstream SL (ASL) performance is given by

$$\mathbb{P}\left(\hat{y}\left(x_{test}; \mathcal{X}_L \cup \mathcal{X}^{s+1}, \mathcal{Y}_L \cup \mathcal{Y}^{s+1}\right) \neq y_{test}\right) - \mathbb{P}\left(\hat{y}\left(x_{test}; \mathcal{X}_L, \mathcal{Y}_L\right) \neq y_{test}\right), \quad (1)$$

where $\hat{y}\left(x_{test}; \cdot\right)$ is the prediction of a classifier trained on data $\cdot$. In contrast, the difference of downstream SSL performance

$$\mathbb{P}\left(\hat{y}\left(x_{test}; \mathcal{X}_L \cup \mathcal{X}_U, \mathcal{Y}_L \cup \mathcal{Y}^{s+1}\right) \neq y_{test}\right) - \mathbb{P}\left(\hat{y}\left(x_{test}; \mathcal{X}_L \cup \mathcal{X}_U, \mathcal{Y}_L\right) \neq y_{test}\right) \quad (2)$$

measures the gain from only newly-acquired labels $\mathcal{Y}^{s+1}$. While ASL is affected by both newly-acquired images and labels, ASSL extricates the two and reveals performance gains from only the newly-acquired labels. The above description motivates one reason to consider ASSL, but it is clear that AL can be applied to improve SSL performance as in (Song et al., 2019).

Despite its importance, we believe two main hurdles restrained prior works to consider ASSL. SSL algorithms have seen great advances only recently (Berthelot et al., 2019; Sohn et al., 2020) and their full strengths simply weren't available. On CIFAR-10, state-of-the-art SSL algorithms presented with as few as 40 labels are now able to match full-supervision where all labeled data is used. Second, SSL performance degrades significantly when the class distribution of labeled data is imbalanced. In AL, a-priori enforcing balanced classes is impossible because labels are unknown and discarding majority classes (under-sampling) wastes what was spent to acquire the labels. When the labeled set's classes are highly imbalanced, pseudo-labels generated by SSL algorithms are even more-so imbalanced (Kim et al., 2020). By adopting a pseudo-label refinement process, we rectify performance degradation caused by class imbalance and are able to achieve increasing performances when incrementally acquiring more labels.

## 2.2 Related Work

### 2.2.1 Active Learning

Only DL-based AL algorithms are surveyed, where version-space approaches become trivial due to their expressive power (Ash et al., 2020). A fully-trained classifier is used to query for labels of samples from a pool of unlabeled data $\mathcal{X}_U$. Many AL algorithms can be characterized by how they valuate each candidate batch $\mathcal{X} \subset \mathcal{X}_U$, where labels corresponding to the batch maximizing some scoring function $v(\mathcal{X})$ are acquired. Among the earliest algorithms, the uncertainty-based algorithms developed in (Wang & Shang, 2014) score each sample $x_i$ using the classifier's margin $v_i = \min_{y' \neq \hat{y}} f_\theta(x_i; \hat{y}) - f_\theta(x_i; y')$, or entropy $H(\sigma(f(x_i; \cdot)))$ where $\sigma$ is the softmax function. Because DNNs are often mis-calibrated and their softmax probabilities are not a good proxy for uncertainty (Guo et al., 2017), a line of work (Kirsch et al., 2019) use Bayesian neural networks (Gal & Ghahramani, 2016). Gissin & Shalev-Shwartz (2019) computes the $\mathcal{H}$-divergence (Ben-David et al., 2010) resulting from hypothetical inclusions of unlabeled samples to the labeled set, and selects those that best aligns the distributions underlying labeled and unlabeled sets. Sener & Savarese (2017) pose each query step as a core-set selection problem and finds an approximate solution. EGL (Huang et al., 2016) and BADGE (Ash et al., 2020) use the gradients of a loss on unlabeled samples as proxies for uncertainty. The former queries for samples that maximize the gradient norm, while the latter diversifies gradient embeddings using k-means++.

### 2.2.2 Semi-supervised Learning

Modern SSL algorithms utilize unlabeled samples and add a consistency loss to act as a regularization in addition to the standard supervision loss. FixMatch (Sohn et al., 2020) is a state-of-the-art algorithm that combines and simplifies a sequence of developed SSL methods (Lee, 2013; Laine & Aila, 2017; Tarvainen & Valpola, 2017; Berthelot et al., 2019) by generating pseudo-labels with weakly-augmented samples. Kim et al. (2020); Lee et al. (2021) observe that pseudo-labels generated by related algorithms (Berthelot et al., 2019; Sohn et al., 2020; Berthelot et al., 2020) are severely imbalanced when the classifier is trained on imbalanced data, thereby degrading performance. In AL, it is impossible to ensure balanced classes in either the labeled or unlabeled sets and the same problem persists. For our problem setting, we use FixMatch-DARP (Kim et al., 2020) where pseudo-labels are post-processed such that their class distribution matches a target distribution. Because for general purposes it is impractical to assume knowledge of class distribution underlying unlabeled samples, we set this target as the uniform distribution.

## 3 Motivations and Method

### 3.1 Notations

A classifier's output layer (preceding softmax) is denoted as $f_\theta$, and the gradients with respect to its parameters as $\nabla f_\theta$. We often drop the subscript $\theta$ and leave it otherwise for emphasis. For simplicity of exposition, we describe our notations assuming a single class and note that this can easily be

re-written following (Arora et al., 2019a; Allen-Zhu et al., 2019; Du et al., 2019) for multi-class classification. Given $N$ samples $\mathcal{X} = \{x_1, \ldots, x_N\}$ and $d$ parameters, the dimension of networks gradient is listed as $\nabla f_\theta(\mathcal{X}) \in \mathbb{R}^{N \times d}$. The Gram matrix $\mathcal{K}_t(\mathcal{X}, \mathcal{X}') := \nabla f_{\theta_t}(\mathcal{X}) \nabla f_{\theta_t}^T(\mathcal{X})$ computed using parameters $\theta_t$ obtained after $t$ optimization (e.g. SGD) steps is also known as the empirical NTK (Arora et al., 2019a).

## 3.2 Motivations

### 3.2.1 Uncertainty Embeddings

Motivated by the ubiquity of stochastic gradient descent (SGD) used to train deep neural networks, Huang et al. (2016); Ash et al. (2020) use gradient embeddings $\nabla \mathcal{L}(\theta; x, \hat{y})$ to measure the uncertainty about a sample $x$ using a proxy label $\hat{y}$. We adopt a similar view on gradients and use them to valuate unlabeled samples, except that our algorithm will make use of the network's gradients $\nabla f$ which is related to the loss gradients $\nabla \mathcal{L}$ through the chain rule. However, while the arguments used in above references are mainly based on the idea that uncertain samples cause large gradients $\nabla \mathcal{L}$, this is not necessarily true for the network's gradients $\nabla f_\theta$. Instead, the gradients' directions are additionally used to measure uncertainty. A network supervised on data including a labeled sample $x_l$ will be more-so certain on that sample than on an unlabeled sample $x_u$, and the product $\nabla f_{\theta_t}(x_l)^T \nabla f_{\theta_t}(x_u)$ being small indicates uncertainty about $x_u$, and in turn that $x_u$ should be queried for its label. We show, after presenting our algorithm in Sec. 3.3, that our selection criterion captures uncertainty information by comparing the gradient's direction with a confident reference vector evaluated at a labeled instance.

### 3.2.2 Batch-mode Operation

Given a fixed labeling budget $|\mathcal{Y}_L| \le B$, a lower bound on the query size $Q$ is determined by how often the classifier can query the label oracle or worker. When the worker is not to be disturbed, a large query size (e.g. $Q = B$) is necessary, and ideally an AL algorithm should attain higher performance when querying more often. To avoid excessive numbers of queries, one of the most important traits of an AL algorithm is batch mode operation, valuating the collective importance of a candidate batch instead of its marginal elements. Early DL-based AL algorithms were myopic $v(\mathcal{X}) = \sum_i v(x_i)$, meaning that their valuation of samples does not consider the collective value of candidate batches. For example, max-margin is a myopic policy and queries redundant samples (Kirsch et al., 2019) when duplicates are present. Algorithms that operate in the batch setting prove critical as query size becomes large.

### 3.2.3 Loss Landscape and Classification Performance

Loss landscape has long been connected to generalization (classification) performance, one view being that critical points near flat minima are more robust to distribution shifts occurring between train and test sets (He et al., 2019). Gradient steps in flat landscapes that do not take into account second order information for re-scaling inevitably take small steps, but it has been observed in (Athiwaratkun et al., 2019) that SGD continues to take large steps when applied to losses used in SSL. Together these views suggest that an improved loss landscape for SSL would enhance generalization performance.

To this end, one of our considerations in designing an AL algorithm is to construct a training set so that the induced landscape exhibits properties positively correlated with generalization as discussed above. We show that in addition to the utilization of uncertainty information from network's gradients and diversity enforcement, another view for our objective is to select data that ameliorates downstream training. Because our algorithm improves the conditioning of downstream optimization problem, we name it Neural Pre-Conditioning (NPC).

## 3.3 Algorithm

Let $\lambda_{\min}(\mathcal{X})$ be the minimum eigenvalue of the symmetric Gram matrix $\mathcal{K}_t(\mathcal{X}, \mathcal{X})$. We propose to encode the uncertainty about samples $\mathcal{X}$ through the network's gradients $\nabla f_{\theta_t}(\mathcal{X})$ used to compute the Gram matrix $\mathcal{K}_t$ and find the subset that solves

$$\max_{\mathcal{X}_u \subset \mathcal{X}_U} \min_{i \le |\mathcal{X}_u \cup \mathcal{X}_L|} \lambda_i(\mathcal{X}_u \cup \mathcal{X}_L). \tag{3}$$

---
**Algorithm 1** Neural Pre-Conditioning (batch-mode solution to (3))
---
Inputs: Unlabeled pool $\mathcal{X}_U$, acquisition size $Q$.
Output: New pool $\mathcal{X}_u^*$ to be labeled.
**for** $i = 1, \cdots, m = \mathcal{O}(N_U)$ **do**
   $\mathcal{X}_u^{(i)} \leftarrow Q$ unlabeled instances randomly sampled from $\mathcal{X}_U$.
   $v(\mathcal{X}_u^{(i)}) \leftarrow \lambda_{\min}\left(\mathcal{X}_L \cup \mathcal{X}_u^{(i)}\right)$ using the network's Gram matrix.
**end for**
Return $\mathcal{X}_u^* \leftarrow \arg\max(v)$

---

To understand how our algorithm makes use of the direction of gradients to embed uncertainty information as argued in Sec. 3.2.1, consider the simple case of selecting one of two unlabeled samples $x_u, x_u'$. Let $\mathcal{X} = \{x_l, x_u, x_u'\}$ be the set containing these two and a labeled sample, and without loss of generality suppose $|\nabla f(x_l)^T \nabla f(x_u)| = a > |\nabla f(x_l)^T \nabla f(x_u')| = b$ with normalized gradients $\|\nabla f(x)\| = 1$ for all $x \in \mathcal{X}$. Given potential labeled sets $X_1 = \{x_l, x_u\}$ and $X_2 = \{x_l, x_u'\}$, the minimum eigenvalues are $\lambda_{\min}(X_1) = 1 - a$ and $\lambda_{\min}(X_2) = 1 - b$. Therefore, NPC measures the model's uncertainty about a sample $x_u'$ by comparing its gradient direction $\nabla f(x_u')$ with a more confident sample's $\nabla f(x_l)$ as a reference, ultimately returning a sample whose gradient direction is further away from the reference's to avoid querying for a less-informative sample.

Next we address how the algorithm enforces diversity for batch-mode queries. Consider multisets $\mathcal{X}$, i.e. $\mathcal{X}$ can have duplicate elements: $\mathcal{X} \neq \mathcal{X} \cup \{x\}$ for any $x \in \mathcal{X}$. A dataset $\mathcal{X}$ with duplicate instances is called degenerate, or equivalently any non-degenerate set $\mathcal{X}$ has elements $\|x_i - x_j\| > 0$ for every pair $i \neq j$ indexing samples in $\mathcal{X}$. We show formally that NPC provably finds only non-degenerate solutions as long as such candidates exist. Proposition 1 alone resolves issues present in many AL algorithms that acquire identical samples on redundant datasets such as "repeated MNIST" (Kirsch et al., 2019).

**Proposition 1** (NPC finds non-degenerate solutions)**.** *Suppose* $x_i \neq x_j \Rightarrow \mathcal{K}^{(T)}(x_i, \cdot) \neq \mathcal{K}^{(T)}(x_j, \cdot)$ *for every* $x_i, x_j \in \mathcal{X}_L \cup \mathcal{X}_U$. *For any degenerate* $\mathcal{X}_u$ *and non-degenerate* $\mathcal{X}_u^*$ *sets,*

$$\lambda_{\min}\left(\mathcal{X}_L \cup \mathcal{X}_u^*\right) > \lambda_{\min}\left(\mathcal{X}_L \cup \mathcal{X}_u\right) = 0. \tag{4}$$

**Remark 1.** *Intuitively, the assumption* $x_i \neq x_j \Rightarrow \mathcal{K}(x_i, \cdot) \neq \mathcal{K}(x_j, \cdot)$ *means that a high dimensional vector (function's gradients) is one-to-one on the small and countable domain* $\mathcal{X}_L \cup \mathcal{X}_U$. *This is true at least in the neighborhood of initialization for ReLU networks as long as not too many neurons are deactivated (Allen-Zhu et al., 2019) or for another class of networks Du et al. (2019).*

*Proof.* The proof is a simple consequence of the rank-nullity theorem and positive definiteness. All eigenvalues computed over non-degenerate sets $\mathcal{X}_L \cup \mathcal{X}_u^*$ are non-zero since row vectors of $\hat{\mathcal{K}}^{(t)}(\mathcal{X}_L \cup \mathcal{X}_u^*)$ are linearly independent. Because $\mathcal{K}$ is semi-positive definite and singular only when its row vectors are linearly dependent, LHS > 0. RHS has duplicate elements in the multiset, and therefore at least two row vectors are linearly dependent. Consequently $\mathcal{K}^{(t)}$ is singular, implying RHS=0. $\qquad\square$

One property that can be inferred from the above proposition is that NPC consolidates labeled data. Interestingly, existing AL algorithms do not explicitly use labeled data when querying labels. Because the labeled set at early acquisition steps may have been constructed using a semi-random acquisition step, or its measurements of uncertainty may have been unreliable because the network had been trained on such few labels, it is important that the label set is also re-evaluated against potential candidates so that label cost is not wasted on nearly-redundant samples' labels.

### 3.4 Computational Considerations

Computing the Gram matrix over a given candidate $\mathcal{X}_u$ requires summing each layer's Gram matrix as $\mathcal{K}_t = \sum_{l=1}^L \mathcal{K}_t^{(l)}$. Because each Gram matrix is semi-positive definite, its minimum eigenvalue is bounded below by the last layer's as $\lambda_{\min}\left(\mathcal{K}_t^{(L)}\right) \geq \lambda_{\min}(\mathcal{K}_t)$. Therefore, we use only the last layer's gradients to compute $\mathcal{K}_t$, where the resulting objective serves as a lower bound to Eq. (3).

Furthermore we replace each block-element whose dimension is the number of classes with its trace to save memory.

When solving the inner-minimization, the kernel's value over labeled samples can be stored and re-used for every candidate batch $\mathcal{X}_u$. We compute the minimum eigenvalue using the robust and efficient locally optimal block preconditioned conjugate gradient method (Stathopoulos & Wu, 2002). However, the search space of Eq. (3) is combinatorial in the pool size and query size. Therefore we approximate the solution to Eq. (3) by sampling $m = \mathcal{O}(N_U)$ subsets uniformly at random to match the runtime of myopic algorithms, where $N_U$ is the unlabeled set's size. By the inclusion-exclusion principle, the top $r N_U$ batches, with $r \in [0, 1]$, are included in the search space with probability $(1 - r)^m$. Taking $m = 1000$ as an example, the randomized search returns a batch within the 99-percentile with probability $\geq 1 - 4 \cdot 10^{-5}$.

## 4  Discussion

### 4.1  A Better Optimization Plateau for Generalization

As motivated earlier, flattening out the landscape has positive implications towards generalization. Here we prove that the landscape induced by labels acquired using NPC allows larger step sizes for convergence, which in turn leads to faster convergence towards flat landscapes. At least for shallow 2-layer networks, increasing the convergence rate also reduces the generalization error (Arora et al., 2019b).

For only this section, assume a non-degenerate training set: $\|x_i - x_j\| > 0$ for each $i \neq j$.

**Theorem 1.** *At each gradient descent iteration $t$ with step size $\eta = \mathcal{O}(\lambda_{\min}(\mathcal{K}_0))$, the MSE loss $\mathcal{L}$ of a properly-initialized, sufficiently wide ReLU network decays as*

$$\mathcal{L}_{t+1} \leq (1 - \mathcal{O}(\eta \lambda_{\min}(\mathcal{K}_t))) \mathcal{L}_t \tag{5}$$

*with high probability over initialization.*

Note that NTK-analyses typically express the training dynamics as a function of $\mathcal{K}_0, \mathcal{K}_\infty$, or the true NTK. Although this can be done with additional perturbation analysis, we leave it at this form since we are concerned with the eigenvalue of the network's Gram matrix.

Two remarks follow. First, the above shows that the set of step-sizes under which gradient descent converges is determined by $\lambda_{\min}(\mathcal{K}_0)$. The kernel $\mathcal{K}_t$ essentially stays constant throughout training for a sufficiently wide network and is fixed as $\mathcal{K}_0$ for simplicity. Therefore, gradient descent can take large step-sizes and still converge when the labeled dataset is constructed using NPC. By maximizing $\lambda_{\min}(\mathcal{K}_\infty)$, where $\mathcal{K}_\infty$ is the Gram matrix of a classifier trained until near-convergence, NPC improves both training and generalization. Second is the withstanding of Thm. 1 when the the computation of $\mathcal{K}$ is reduced by using the last layer's Gram matrix. We described in Sec. 3.4 that our NPC algorithm solves Eq. (3) by replacing $\mathcal{K}_t$ in $\lambda_{\min}(\mathcal{K}_t)$ with the last layer's Gram matrix. As shown, training and generalization benefits that come from solving Eq. (3) still hold when using the network's last layer to compute the kernel.

### 4.2  Benefits of Randomized Search

The alternation between querying for labels and training can be interpreted as a feedback system, which illustrates the exploration vs. exploitation effect of randomization used to solve Eq. (3) and complements the view that the network's uncertainty about samples is minimized with more labels. A state described by trained parameters $\theta$ and training set $\mathcal{Z}_L, \mathcal{X}_U$ is used by a policy, which selects $\mathcal{X}_u^*$ and acquires (observes) $\mathcal{Y}_u^*$. At the first acquisition step, the network's gradients are unreliable measures of uncertainty embeddings due to a lack of labeled samples. Subsequent states are then updated by propagating newly acquired labels to train the network so that gradient embeddings better represent uncertainty about samples. At early acquisition stages, the randomized search therefore encourages the acquisition policy to explore instead of relying excessively on its belief. The search space size decreases with more acquisition steps and therefore the policy progressively exploits its belief.

Our algorithm's effect on generalization error can also be understood by studying the infinite-width regime, where we view randomization to act as a regularization method considering that we use finite-

Table 1: CIFAR-10 ($Q_0 = 10, Q = 20$): Average accuracy (%) $\pm$ standard deviation. Initial model achieved $57.64\%$ accuracy.

| # Labels
Algorithm | 30 | 50 | 70 |
|---|---|---|---|
| Passive | $78.08 \pm 5.48$ | $91.82 \pm 2.30$ | $91.00 \pm 2.78$ |
| Margin | $87.36 \pm 5.01$ | $90.85 \pm 3.64$ | $90.85 \pm 3.64$ |
| ALBL | $80.61 \pm 12.5$ | $89.62 \pm 6.66$ | $94.45 \pm 0.20$ |
| BADGE | $80.60 \pm 4.46$ | $86.43 \pm 1.22$ | $80.32 \pm 7.76$ |
| NPC$^{\dagger}$ | $85.09 \pm 9.61$ | $\mathbf{94.63 \pm 0.07}$ | $\mathbf{94.85 \pm 0.02}$ |

width networks. A finite-width network's prediction is approximately kernel ridgeless regression (Arora et al., 2019a). Bordelon et al. (2021) showed that for kernel ridgeless regression, a training point reduces generalization error at modes corresponding to larger eigenvalues. Our objective in Eq. (3) selects a training set that maximizes the minimum eigenvalue, and therefore enhances data efficiency in the sense that generalization error is affected at as many modes as possible. However, finite-width ConvNets are generally better classifiers (Arora et al., 2020; Lee et al., 2020) which were therefore used to query for labels. The eigen-spectrum of a finite width network's Gram matrix is not identical to the NTK and solving the objective exactly may not directly translate to more generalization modes as for infinite width networks.

## 5 Experiments

### 5.1 Implementation Details

We adopt all SSL-related configurations from (Oliver et al., 2018) and use the WRN-28-2 architecture (Zagoruyko & Komodakis, 2016) for all experiments. At the first acquisition step, we randomly sampled 1 image per class and used the model that attained median performance across 5 trials. Subsequent acquisitions were performed with query size $Q = 20$ for CIFAR-10 and $Q = 200$ for CIFAR-100. All performances are averaged over 3 trials. Following most AL setups, we train classifiers from scratch after each acquisition step. Training from scratch better assesses the value of labels as it mitigates the possibility of vicious cycles where models trained sub-optimally in previous acquisition steps have no hope of improving despite superb data.

As discussed earlier, we assume no a-priori information on class distribution underlying unlabeled data. Instead of estimating the underlying class distribution as done by Kim et al. (2020) which may be detrimental given few labels, we simply take the target pseudo-label distribution to be uniform and perform pseudo-label refinement accordingly.

### 5.2 Baseline Algorithms

The proposed NPC algorithm is compared with passive learning where labels are queried uniformly at random, margin (Roth & Small, 2006), active learning by learning (ALBL, Hsu & Lin (2015)) comprising least confidence and Coreset (Sener & Savarese, 2017), and BADGE. Margin evaluates the classifier's margin and selects $Q$ samples whose margin is lowest. ALBL employs a two-armed adversarial bandit algorithm to adapt to the better of least confidence $\arg\min_i f(x_i)$ and Coreset. BADGE was described earlier, and acquires samples by applying k-means++ on gradient embeddings. Entropy was also considered but excluded because of its low performance on some experiments.

### 5.3 Performance

Tables 1 and 2 show the accuracy of AL algorithms when trained on CIFAR-10 and CIFAR-100, respectively. NPC outperforms other label acquisition schemes on nearly all dataset sizes and is at least competitive on the few others. Although BADGE is state-of-the-art on AL benchmarks, we observe older algorithms performing better when evaluated by SSL accuracy. This reveals how existing AL algorithms have been evaluated by their efficiency of sample acquisitions rather than label complexity.

Table 2: CIFAR-100 ($Q_0 = 100, Q = 200$): Average accuracy (%) $\pm$ standard deviation. Initial model achieved 21.29% accuracy.

| # Labels
Algorithm | 300 | 500 | 700 | 900 |
|---|---|---|---|---|
| Passive | $37.98 \pm 1.89$ | $47.11 \pm 1.41$ | $52.69 \pm 1.16$ | $57.97 \pm 1.43$ |
| Margin | $37.41 \pm 1.22$ | $48.21 \pm 3.76$ | $52.53 \pm 0.75$ | $57.03 \pm 0.39$ |
| ALBL | $39.05 \pm 0.62$ | $\mathbf{49.88 \pm 0.92}$ | $54.23 \pm 1.78$ | $56.65 \pm 0.86$ |
| BADGE | $24.55 \pm 1.13$ | $25.21 \pm 1.86$ | $28.03 \pm 1.74$ | $29.81 \pm 1.72$ |
| NPC$^\dagger$ | $\mathbf{40.92 \pm 1.65}$ | $48.16 \pm 0.77$ | $\mathbf{55.76 \pm 1.45}$ | $\mathbf{58.58 \pm 0.52}$ |

To complement performances, Fig. 1 illustrates the similarity between algorithms as the intersection over union (IoU) of label indices as more labels are collected on CIFAR-100. At any given label set size $N_L$, an algorithm's label set is the union of labels acquired at different trials. Margin and least-confidence both rely heavily on the classifier's predictions, and labels acquired at different trials overlap significantly. As shown, ALBL and Margin are similar in how labels acquired, which describes that using a classifier's least confidence is similar to acquiring based on its margin. On the other hand, other pairs of algorithms have very small overlaps, demonstrating that their acquisition criteria are drastically different.

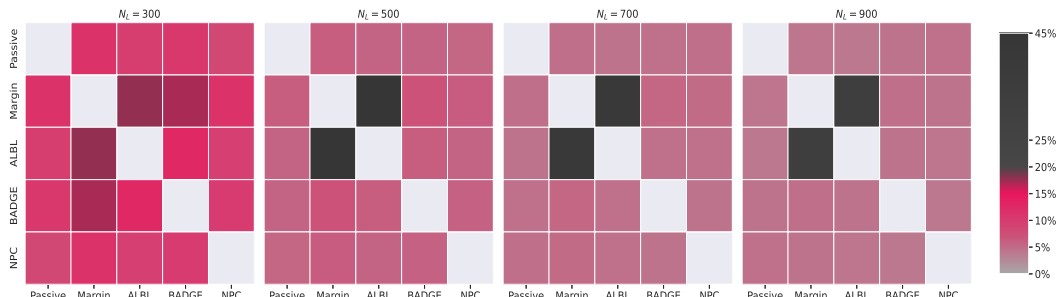

Figure 1: Similarity between AL algorithms: Intersection over union (IoU%) of labels commonly acquired by algorithms.

## 5.4 Reducing Inquiry Frequency

The above experiments aim to maximize accuracy with budget constraints on the number of labels. Certain applications may additionally require that the number of inquiries is minimized to reduce the frequency of interaction between classifier and annotator. It is clear that RANDOM remains unaffected by the number of inquiries, and it is desirable that AL algorithms maintain high performance when fewer inquiries are possible.

As observed in Tab. 1, NPC achieves very high performance at $N_L = 50$. To accommodate a limit on the number of inquiries, we experiment with how NPC and BADGE, selected based on their similarity, are affected on what we call single and zero shot AL, both referring to a single query given a model trained on very few labels ($N_L = 10$) and a randomly initialized ($N_L = 0$) model, respectively. As shown in Tab. 3, NPC excels in both zero and single shot settings where an imperfect classifier is used to valuate samples. Interestingly, NPC remains nearly unaffected by which model is used to query for samples in zero or single shot queries. In contrast, BADGE is detrimentally affected by its over-reliance on gradient embeddings on single-shot AL and rather performs better in zero-shot queries when gradients are randomly initialized.

A few questions arise from this observation. The fact that BADGE performs worse on single-shot AL with a larger query size highlights that gradients alone may not be informative features in valuating samples as assumed. NPC also uses gradients as features for valuating samples, but its robust performance with respect to number of queries can be attributed to our conclusion from theoretical analysis where problem conditioning is directly affected.

Table 3: Zero-shot and Single-shot AL on CIFAR 10 using $Q \in \{30, 50\}$, where zero-shot refers to acquisition using a randomly initialized model and single-shot to a model trained on 1 label per class.

| | Zero Shot | | Single Shot | |
|---|---|---|---|---|
| # Labels | 40 | 60 | 40 | 60 |
| BADGE | $92.13 \pm 3.24$ | $94.42 \pm 0.41$ | $86.70 \pm 5.42$ | $70.28 \pm 13.22$ |
| NPC$^\dagger$ | $93.14 \pm 1.42$ | $93.16 \pm 2.34$ | $92.26 \pm 3.26$ | $93.76 \pm 0.58$ |

It's surprising how both BADGE and NPC perform extremely well on zero-shot AL, where a randomly initialized model decides which labels are most valuable. For comparison, FixMatch on *balanced data* without DARP reportedly achieves $86.19\%$, comparable to BADGE on single-shot but under-performing both BADGE and NPC on zero-shot. To explain this phenomenon, it is instructive to view randomly initialized networks in their asymptotic limits.

At first glance it may appear that NPC with a randomly initialized network should not work well. However, wide networks at initialization approximate their infinite-width NTK (Arora et al., 2019a). As mentioned earlier, fully-trained wide networks are essentially ridge regression $\hat{y}_{ridge}(x_{test}) = \mathcal{K}(x_{test}, X)\mathcal{K}^{-1}(X, X)y$, and zero-shot NPC translates to a construction of the above kernel on which ridge regression will be performed. A training point influences the generalization of kernel regression more for modes corresponding to large eigenvalues (Bordelon et al., 2021), which is maximized by NPC. In summary, NPC using a randomly initialized network selects samples to maximize generalization performance as predicted by approximate kernel regression through the NTK spectrum.

# 6 Conclusion

This work motivated downstream SSL performance as a benchmark to evaluate AL algorithms. We then described motivations recurrent in previous works and proposed an AL algorithm that addresses these concerns. The proposed NPC algorithm captures uncertainty through the model's gradients, operates in the batch-mode setting, and improves the landscape of downstream SSL through data acquisition as measured by properties related to generalization. Experiments re-evaluating state-of-the-art AL algorithms with respect to downstream SSL performance, which better measures label complexity, demonstrate that NPC outperforms other AL algorithms on most dataset sizes and tasks.

NPC enjoys several properties that aren't present in other AL algorithms or is at least not obvious. First, NPC explicitly consolidates existing labeled data when measuring the value of labeling un-labeled candidates. Moreover, NPC is a batch AL algorithm that provably selects distinct samples. The proposed algorithm is also interesting in that it is a kernel-based sampling scheme. Kernels are excellent models of data distributions, and NPC's construction of a kernel using DL opens new venues for AL.

A few limitations and future works are described. Our experiments rely on modern SSL algorithms to evaluate AL algorithms. Although current SSL algorithms achieve extremely high accuracy on vision tasks, they suffer from algorithmic instability where given the same model and dataset, their performances vary more-so than supervised learning. Ideally, all algorithms should achieve higher accuracy in line with the "more data is better" principle. Because performance deterred by class imbalance is resolved using pseudo label refinements, we believe experimental evaluations will benefit most from algorithmic stability. Further, SSL training demands much more computation than SL counterparts, and consequently an exhaustive evaluation of various AL algorithms is prohibitive. Experimental protocols that reduce computations in evaluating algorithms yet are fair would expedite research. Lastly, we treat AL and SSL phases independently for our purpose. An interesting direction to pursue would be to design AL and SSL schemes that adapt to each other. For example, our theoretical analysis and NPC's valuation gives an upper bound on possible learning rates for downstream training. By designing learning schedules to adapt to the set of admissable step sizes, downstream training may be better stabilized and achieve higher performance.

## Acknowledgments and Disclosure of Funding

We thank Professor R. Srikant for helpful discussions relating to Neural Tangent Kernels.

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
