# Appendix: A Neural Pre-Condition Active Learning Algorithm to Reduce Label Complexity

## 1   A   Proof of Theorem 1

2 Assume a non-degenerate training set $\|x_i - x_j\| > 0, \forall i \neq j$. Theorem 1 in the main script is
3 re-written:

4 **Theorem 1.** *At each gradient descent iteration $t$ with step size $\eta = \mathcal{O}(\lambda_{\min}(\mathcal{K}_0))$, the MSE loss $\mathcal{L}$*
5 *suffered by a properly-initialized feedforward ReLU network decays as*

$$\mathcal{L}_{t+1} \leq (1 - \mathcal{O}(\eta \lambda_{\min}(\mathcal{K}_t))) \mathcal{L}_t \qquad (1)$$

6 *with high probability over initialization.*

7 We adopt the convention that all gradients are flattened in vector form and use the Euclidean norms
8 to represent their size. First we express training dynamics as a recursion:

9 **Lemma 1.** *Feedforward DNNs with once-differentiable activation functions trained using gradient*
10 *descent on the MSE loss $\mathcal{L}_t$ with step size $\eta$ follows the recursion:*

$$\mathcal{L}_{t+1} \leq (1 - \eta \lambda_{\min}(\mathcal{K}_t)) \mathcal{L}_t + \xi_t + \epsilon_t, \qquad (2)$$

11 *where $\xi_t = \int_0^\eta \nabla \mathcal{L}_t^T (\nabla \mathcal{L}_t - \nabla \mathcal{L}(\theta_t - \gamma \nabla \mathcal{L}_t)) d\gamma$ and $\epsilon_t = \frac{1}{2}(f_{\theta_{t+1}} - f_{\theta_t})^2$.*

12 *Proof.* This derivation is mostly from Du et al. (2019), but we include the proof under our notations
13 for completeness. Let $e_t = y - f_{\theta_t}$. A standard technique with triangular inequality gives

$$\mathcal{L}_{t+1} \leq \mathcal{L}_t + \|f_{\theta_{t+1}} - f_{\theta_t}\|^2 - 2e_t^T \left(f_{\theta_{t+1}} - f_{\theta_t}\right). \qquad (3)$$

14 Let $h(\eta) = f(\theta_t - \eta \nabla \mathcal{L}_t)$. By the fundamental theorem of calculus,

$$f_{\theta_{t+1}} - f(\theta_t) = h(\eta) - h(0)$$

$$= \int_0^\eta h'(\gamma) d\gamma = \int_0^\eta h'(0) d\gamma + \int_0^\eta h'(\gamma) - h'(0) d\gamma$$

15 Since $h'(0) = -\nabla f(\theta_t)^T \nabla \mathcal{L}_t = -e \nabla f_{\theta_t}^T \nabla f_{\theta_t} = -e \text{Tr}(\mathcal{K}_t)$, we have

$$e^T(f_{\theta_{t+1}} - f_{\theta_t}) = -\eta e^T \mathcal{K}_t e + \int_0^\eta h'(\gamma) - h'(0) d\gamma \leq -\eta \lambda_{\min}(\mathcal{K}_t) \mathcal{L}_t + \xi_t.$$

16 Substituting into Eq. 3 gives Eq. 2 together with $e_t \int_0^\eta h'(\gamma) - h'(0) d\gamma = $
17 $\int_0^\eta \nabla \mathcal{L}_t^T (\nabla \mathcal{L}_t - \nabla \mathcal{L}(\theta_t - \gamma \nabla \mathcal{L}_t)) d\gamma$. $\qquad \square$

18 The above bound sheds light on training dynamics, where the first term decreases linearly with rate
19 determined by the Gram matrix' eigenvalue. To establish Thm. 1 that states the loss descends at each
20 gradient step, it remains to prove that residual terms $\xi_t, \epsilon_t$ grow (sub-)linearly with $\mathcal{L}_t$.

21 An extension of smoothness and convexity is defined following (Allen-Zhu et al., 2019):

**Definition 1** (Smoothness). *A non-negative, once-differentiable function $g \in C^1(\mathcal{X})$ is $(\alpha, \beta)$-smooth if for every $x, y \in \mathcal{X}$,*

$$g(y) \leq g(x) + \nabla g(x)^T (y - x) + \alpha \sqrt{g(x)} \|y - x\| + \beta \|y - x\|^2 \tag{4}$$

**Definition 2** (Near-Convexity). *A non-negative function $g \in C^1(\mathcal{X})$ has gradients $\nabla g$ that scale as $(\mu, M)$ if*

$$\mu g(x) \leq \|\nabla g(x)\|^2 \leq M g(x), \forall x \in \mathcal{X}. \tag{5}$$

*If a function's gradients scale as $(\mu, M)$, we say the gradient scale is bounded.*

First we invoke the following lemma (Thms. 3 & 4 in Allen-Zhu et al. (2019)) to show that the MSE loss remains semi-smooth and nearly convex throughout training for wide ReLU networks:

**Lemma 2.** *For sufficiently small $\|\theta - \theta_0\|$ and $\|\theta - \theta'\|$, the loss remains nearly convex*

$$\|\nabla \mathcal{L}(\theta)\|^2 = \Theta(\mathcal{L}(\theta))$$

*and semi-smooth*

$$\mathcal{L}(\theta') \leq \mathcal{L}(\theta) + \nabla \mathcal{L}(\theta)(\theta' - \theta) + \mathcal{O}\left(\mathcal{L}(\theta)^{1/2} \|\theta' - \theta\|\right) + \mathcal{O}\left(\|\theta' - \theta\|^2\right)$$

*with high probability hiding constants depending on architecture width, depth, and dataset size.*

Above we use $\Theta(\cdot)$ as upper and lower bounds matching up to multiplicative constants.

Next we bound the residual terms in Lemma 1:

**Lemma 3.** *If the loss function $\mathcal{L}_t$ remains smooth and near-convex as defined above,*

$$\epsilon_t, \xi_t \leq \mathcal{O}(\eta^2) \mathcal{L}_t$$

*with high probability over initialization.*

*Proof.* The following inequality will be used for $(\alpha, \beta)$-smooth functions.

**Proposition 1.** *If $g$ is $(\alpha, \beta)$-smooth,*

$$(\nabla g(y) - \nabla g(x))(y - x) \leq \alpha(\sqrt{g(x)} + \sqrt{g(y)})\|y - x\| + 2\beta \|y - x\|^2 \tag{6}$$

*Proof.* Expanding the LHS in terms of $x$ and $y$ then summing their upper bounds gives the inequality. $\square$

**Bound on $\xi_t$** Proposition 1 with $\mathcal{L}$ at $\theta_t$ and $\theta_t - \gamma \nabla \mathcal{L}_t$ can be used to bound the integrand.

$$(\nabla \mathcal{L}_t - \nabla \mathcal{L}(\theta_t - \gamma \nabla \mathcal{L}_t)) \nabla \mathcal{L}_t \leq \alpha \|\nabla \mathcal{L}_t\| \left(\sqrt{\mathcal{L}_t} + \sqrt{\mathcal{L}(\theta_t - \gamma \nabla \mathcal{L}_t)}\right) + 2\gamma \beta \|\nabla \mathcal{L}_t\|^2 \quad .$$

Using the definition of smoothness

$$\mathcal{L}(\theta_t - \gamma \nabla \mathcal{L}_t) \leq \mathcal{L}_t + \gamma \left(\alpha \sqrt{\mathcal{L}_t} \|\nabla \mathcal{L}_t\| - \|\nabla \mathcal{L}_t\|^2\right) + \beta \gamma^2 \|\nabla \mathcal{L}_t\|^2 \quad ,$$

and by near-convexity,

$$\leq \left(1 + \gamma(\alpha \sqrt{M} - \mu) + \beta \gamma^2\right) \mathcal{L}_t. \tag{7}$$

Let $b = \left(\alpha \sqrt{M} - \mu\right)/2\beta$ and $c = 1/\beta - b^2$.

$$\sqrt{\mathcal{L}_t} + \sqrt{\mathcal{L}(\theta_t - \gamma \nabla \mathcal{L}_t)} \leq \sqrt{\mathcal{L}_t}\left(1 + \sqrt{\beta}\left(\gamma + |b| + \sqrt{|c|}\right)\right) =: \sqrt{\mathcal{L}_t}\left(\sqrt{\beta}\gamma + c'\right)$$

by the triangle inequality. Again, $\|\nabla \mathcal{L}_t\|^2 \leq M \mathcal{L}_t$, and we have a bound on the integrand as

$$\alpha \|\nabla \mathcal{L}_t\| \left(\sqrt{\mathcal{L}_t} + \sqrt{\mathcal{L}(\theta_t - \gamma \nabla \mathcal{L}_t)}\right) + 2\gamma \beta \|\nabla \mathcal{L}_t\|^2 \leq \left(\alpha \sqrt{M}\left(\sqrt{\beta}\gamma + c'\right) + 2\gamma \beta M\right) \mathcal{L}_t$$

$$=: \left(a'\gamma + c''\right) \mathcal{L}_t$$

$$\Rightarrow \xi_t \leq \mathcal{L}_t \int_0^\eta a'\gamma + c'' d\gamma = O\left(\eta^2\right) \mathcal{L}_t.$$

45    where we hide constants that depend on the architecture and dataset size.

46    **Bound on** $\epsilon_t$ It is sufficient that $\epsilon_t \leq \left(a\eta^2 + \lambda_{\min}\eta\right)\mathcal{L}_t$ for any $a$ so that $\mathcal{L}_t$ is guaranteed to decrease
47    for small $\eta$. This proof is quite involved and relies on analytic expressions for ReLU networks. To
48    this end, we follow the setting in Allen-Zhu et al. (2019) and WLOG fix the last layer's weights as $B$,
49    denoting pre- and post- activations by $g^l, h^l$ respectively and an 'active-indicator' matrix $D^l \in \mathbb{R}^{d \times d}$,
50    $D^l_{k,k} = \mathbf{1}\left\{g^l_{k,k} \geq 0\right\}$, and weight matrices $W_l \in \mathbb{R}^{d \times d}$ for each layer $l \in [L]$, where $d$ denotes the
51    width of the hidden layers and $L$ is the number of layers.

52    Notice that for ReLU networks, we can write the post-activations at every layer as $h^l_{t+1} - h^l_t = $
53    $D^l_{t+1}W_{t+1}h^{l-1}_{t+1} - D^l_t W^l_t h^{l-1}_t$.

54    **Proposition 2** (Distributive diagonal matrices)**.** *There exists* $\tilde{D} = \left(\tilde{D}^1, \ldots, \tilde{D}^L\right)$ *with* $\tilde{D}^l \in$
55    $[-1,1]^{d \times d}$ *for every* $l$ *such that*

$$D^l_{t+1}W^l_{t+1}h^l_{t+1} - D^l_t W^l_t h^{l-1}_t = \left(D^l_t + \tilde{D}^l\right)\left(W^l_{t+1}h^{l-1}_{t+1} - W^l_t h^{l-1}_t\right).$$

56    The above proposition follows from case-by-case considerations of ReLU activations, see Proposition
57    11.3 in Allen-Zhu et al. (2019).

58    **Proposition 3** (Linear expansion of post-activations)**.** *There exists some* $\tilde{D}^l \in [-1,1]^{d \times d}$ *at each* $l$
59    *such that*

$$h^l_{t+1} - h^l_t = -\eta \sum_{r=1}^{l}\left(D^l_t + \tilde{D}^l\right)W^l_t \cdots W^{r+1}_t \left(D^r_t + \tilde{D}^r\right) \times \left(\nabla_{W^r_t}\mathcal{L}_t\right)h^{r-1}_{t+1}$$

60    The following proposition due to Allen-Zhu et al. (2019) (Lemma 8.6b and Lemma 7.1, respectively)
61    gives bounds on the first line on the RHS and last term:

62    **Proposition 4.** *For every* $l \in [L]$ *and* $r \in [l]$,

$$\left\|\left(D^l_t + \tilde{D}^l\right)W^l_t \cdots W^{r+1}_t \left(D^r_t + \tilde{D}^r\right)\right\| \leq O(\sqrt{L})\|h^{r-1}_{t+1}\| \leq o(1).$$

63    Applying Cauchy-Schwartz inequality and the fact that norm of sums $\leq$ sum of norms to Propositions
64    2 and 3,

$$\|f_{\theta_{t+1}} - f_{\theta_t}\| = \|B\left(h^L_{t+1} - h^L_t\right)\|, \leq \eta O(L^{1.5}\sqrt{d})\|\nabla\mathcal{L}_t\|.$$

65    Since $\|\nabla\mathcal{L}_t\| \leq \sqrt{M\mathcal{L}_t}$,

$$\epsilon_t = \|f_{\theta_{t+1}} - f_{\theta_t}\|^2 \leq O(L^3 dM)\eta^2 \mathcal{L}_t = O(\eta^2)\mathcal{L}_t. \tag{8}$$

66    $\square$

67    Theorem 1 is a direct consequence of Lemmas 1 and 3, and the step-size can be selected based on $\mathcal{K}_0$
68    because $\mathcal{K}_t$ remains in a neighborhood of $\mathcal{K}_0$ throughout training (Arora et al., 2019).