# OpenReview forum: "A Neural Pre-Conditioning Active Learning Algorithm to Reduce Label Complexity"
_NeurIPS.cc/2022/Conference — NeurIPS 2022 Accept_

### Official Review · Reviewer_96bx · 2022-06-19

**Rating:** 4
**Confidence:** 4
**Soundness:** 2 fair
**Presentation:** 3 good
**Contribution:** 2 fair

**Summary:**

This work proposes a batch mode active learning method for active learning. The active sampling criterion mainly leverages the gradient of the currently trained model with respect to both labeled and unlabeled data to sample the informative data batch for humans to label. A theorem is presented to show that if the proposed sampling strategy is ideally implemented, the model loss decay rate is subject to linear.

**Questions:**

I am interested in how the label information omits in the loss decay bound in theorem1. Can you explain how the inequality in appendix line 16 is derived? I think that is where the label information disappeared in the loss decay.

**Strengths And Weaknesses:**

Strength:
The methodology section is well organized; the proposed method is clearly described and explained with the underlying motivation, making it easy to read and understand.
Weakness:
1. The proposed method can be seen as a variation of the gradient embedding-based active learning while the main difference is the gradient of the model output, rather than the loss function, is used to construct the active sampling function. However, I am not convinced that the uncertainty can be fully captured without considering the potential labels an unlabeled data instance could have during the sampling. The paper mentioned in multiple places that the gradient of f(x) contains the uncertainty of x. But the paper does not spare any effort to prove/demonstrate that. In fact, I believe the label-instance pair is the menial unit for informative measurement in active learning. Two identical data points with different labels(provided by a human annotator) could have significantly different empacts on the model training. But the proposed sampling method treats them as the same. I do not believe the proposed method can perform uncertainty sampling as claimed in the paper.

2. The batch mode sampling is impractical to me because eq3 suggests that the optimal batch needs to be searched within a solution space of combination(N, Q) where N is the pool size and Q is the batch size. For a practical active learning problem, N can easily exceed 10000, and Q is typically set from 50~500, making combination(N, Q) unreachable. The proposed batch solution in algorithm 1 only randomly tests m batches out of combination(N, Q)  solutions. It is almost impossible to have the optimal batch in that m test batches. No need to mention that each batch testing has O((T+Q)^3) complexity given the T is the current labeled data.

3. The bound of the decay rate proposed in theorem1 is based on the optimal batches. I don’t think it will hold for most of the time since the actual batch mode sampling could not guarantee finding the optimal batch.

---

> ### Author Response · Authors · 2022-08-01
> **Author Response to Reviewer 96bx**
>
> Thank you very much for your thoughtful and constructive comments.
>
> We tried to address your comments on the probelm setting, finding optimal batch, and the bound of the decay rate.
>
> ---
> # Weakness
> ### 1. Regarding Problem Setting
> - There is an important difference between the proposed NPC algorithm and existing active learning algorithms in addition to using the gradient of the model output: Our NPC algorithm embeds pairwise comparisons as an entry to the Gram matrix, whose minimum eigenvalue is then used to enforce diversity, by taking the inner product between gradients. We would like to clarify that we take the common assumption in active learning that the human annotator is perfect and therefore labels corresponding to an identical sample cannot be different. This will be clearly described in our next revision. In addition to this noise-free setting, NPC does consider potential labels as the gradients of the model’s outputs (predictions over all classes) are used.
>
> ### 2. Regarding Finding Optimial Batch
> - The reviewer is correct in understanding that the optimal batch cannot be found in a reasonable time for practical purposes. We described this issue and how a randomized search provides an efficient trade-off between optimality and computation in lines 229-233; the top-$r$ percentile batch is included within m randomly selected subsets with probability $(1-r)^m$. As a simple numerical example, we show that the top 1% batch can be found with very high probability by searching through 1000 random subsets instead of all subsets.
>
> ### 3. Regarding Decay Rate
> - The decay rate in Theorem 1 is not based on what would have been the optimal batch, but is a function of any candidate batch and therefore holds for any batch whenever included in the labeled set. This decay rate however is maximized by the (nearly) optimal batch.
>
> ---
>
> # Questions
> ### Regarding Label Information
> - The label information is, as the reviewer carefully noticed, included in appendix line 16 but the notation omits this. To elaborate, $e = y - f$ is a function of the label $y$. Given this noise-free setting, both Huang et al. (2016) and Ash et al. (2020) are based on gradients capturing uncertainty, motivated by references therein, and we also describe in Sec. 3.3 how gradients capture uncertainty. As an example described, we take the fully-trained model’s gradient at a labeled point as a reference vector (row index). The row contains comparisons with gradients evaluated at other unlabeled pairs, and this row vector is low-dimensional if unlabeled samples are similar to labeled samples (less diversity). Inner product between the gradient at a labeled point and gradients evaluated at uncertain (e.g. highly stochastic) unlabeled datapoints will have a large variance, and the minimum eigenvalue of the full Gram matrix will be relatively large. An extreme case is given in Proposition 1, which describes how NPC avoids selecting degenerate (e.g. already-labeled) data.

---

> > ### Comment · Reviewer_96bx · 2022-08-10
> > **Thank you for the response**
> >
> > For Q1, I now agree that the proposed method could capture the uncertainty given the perfect annotator assumption.
> > For Q2, my concern remains, because it seems that the result of ''top 1% batch can be found with very high probability by searching through 1000 random subsets instead of all subsets.'' may not be guaranteed for other tasks and for some tasks even running through 1000 random subsets could be time-consuming.

---

> > > ### Author Response · Authors · 2022-08-10
> > > **Search time matches fastest AL algorithms**
> > >
> > > Thank you for the continued discussion. We proposed to search through O(n) random subsets, where n is the number of individual samples, to match the runtime of myopic AL algorithms whose runtime cannot be improved upon. Therefore, while it is true that searching through 1000 (or O(n)) random subsets can be time-consuming depending on the application, this is a problem for all AL algorithms.

---

### Official Review · Reviewer_h4Gx · 2022-07-11

**Rating:** 6
**Confidence:** 4
**Soundness:** 3 good
**Presentation:** 3 good
**Contribution:** 3 good

**Summary:**

Semi-supervised learning and activa learning are two representative learning paradigms to learn with incomplete supervised data. The difference lies in that active learning relies on human knowledge. This paper argues that the enhancements of SSL performance could be used to evaluate active learning methods to measure their label efficiency. This is an interesting idea. The proposed method is based on a neural tangent kernel analysis, which evaluates unlabeled data based on how they would contribute to updating the training set's inclusion. This is technically sound. The experiments also demonstrate the effectiveness of the proposal.

**Questions:**

1) Can the proposal be applied to more realistic SSL scenarios? For example, recently there have been many studies about open-set SSL and class-imbalanced SSL. Can the proposal be applied with these methods?

**Limitations:**

Yes

**Strengths And Weaknesses:**

Strengths:

1) The paper gives a new perspective on semi-supervised learning and active learning, and shows that the enhancements in SSL performance could be used to evaluate active learning methods. This is a new idea and could inspire subsequent researchers.

2) The proposed method is technically sound.

Weakness

1. The experiments could be further improved. For example, conduct experiments in more settings, such as imbalanced semi-supervised learning and open-set semi-supervised learning. Moreover, it would be better to report performance on more related methods.

---

> ### Author Response · Authors · 2022-08-01
> **Author Response to Reviewer h4Gx**
>
> Thank you very much for your positive feedback and constructive comments.
>
> We tried to answer to your comments and question as follows:
>
> ---
> # Weakness
> ### Regarding Additional Experiments
> - We thank the reviewer for suggesting additional experimental settings for the proposed method. All our experiments are conducted in the more practical, imbalanced semi-supervised learning setting. We believe that designing an active learning for the open-set semi-supervised learning is a whole new problem as this would require modifying existing AL algorithms significantly. For example, if an active learning algorithm queries for novel category labels should we treat this as an undesirable trait? If not, how should we modify our or existing algorithms to use these novel category labels for future queries to improve performance? We agree however that this would be an interesting problem for future work.
>
> ---
>
> # Questions
> ### Regarding the Applicability to Open-set and Class-imbalanced SSL
> - As above, our proposed method is shown to improve performance in class-imbalanced SSL, i.e. the setting considered by Kim et al. (2020). While it is not immediately clear how to extend our AL algorithm to the open-set SSL setting, it would be an interesting problem to consider for future research.

---

### Official Review · Reviewer_yqNP · 2022-07-14

**Rating:** 6
**Confidence:** 3
**Soundness:** 3 good
**Presentation:** 2 fair
**Contribution:** 3 good

**Summary:**

This submission proposes the neural pre-conditioning (NPC) algorithm to leverage the unlabelled data using semi-supervised learning and acquire the valuable data that could be labelled carefully active learning strategy. Furthermore, this work mentioned that traditional active learning methods are typically evaluated with the supervised learning method, which cannot reflect how much the trained model gains the performance from newly acquired labels. Instead, they argue that semi-supervised learning can be a benchmark method to evaluate the Active learning downstream task. The authors state how the NPC algorithm uses the direction of gradients to select the uncertainty samples and prove that the batch-mode operation can increase the diversity of the data selection. They extend the experiments on two datasets with different query sizes and the number of labelled data, comparing the NPC against the other four state-of-the-art active learning methods, which shows the NPC achieves exceptionally high accuracy. In the end, they concluded this work and discussed the limitations and future study directions.

**Questions:**

1. What’s the performance of showing in Fig.1?
2. In the implementation part (the 288th  row), how did you randomly sample one image per class? OR that is the preprocess of selecting the labeled training data?
3. The augment of “SL fails to extricate performance gains from newly acquired labels from the influence of newly acquired samples. In contrast, SSL trains a model on Z and the only difference in datasets used for training is the label, and performance improvements between rounds s and s + 1 reveal gains from newly acquired labels” is the key point in this paper. Intuitively, the data and label act simultaneously to improve the model's performance. Why did you mention that semi-supervised learning is better for evaluating the effect of the newly acquired labels? Could you please give more explanation?


**Strengths And Weaknesses:**

Strengths
1. The results have significant improvement compared to the previous work, achieving state-of-the-art results.
2. Quality: This work reads as a complete paper that has complete sections and clear formatting. There are extended comparative experiments to evaluate the proposed method's effectiveness.

Weaknesses
1. The notations and some presentations are unclear. For example, in eq.1, χ*, Y*and Y_U; in Table 1. What is the Q_0; in 277th row, -u.
2. Clarity: The presentation is not clear in some parts, like in the introduction part, you mentioned that simply replacing the SL with SSL would introduce a new problem because the class imbalanced issue can deteriorate the SSL performance significantly. Although here you added the reference, I think that supplying more details that explain why this issue induces the SSL performance so significant is necessary since the imbalance issue would also degrade the SL.
3. Clarity: For the algorithm section, I consider giving more proof in detail is better. The analysis of the two main contributions on the algorithm, gradients direction and diversity, is not very clear.
4. Originality: this work builds upon prior solutions to improve the performance of active learning using semi-supervised learning in the downstream tasks. However, the novelty is limited.

---

> ### Author Response · Authors · 2022-08-01
> **Author Response to Reviewer yqNP**
>
> Thank you very much for your positive feedback and constructive comments.
>
> We tried to address your comments on the notation, clarity and originality and answered to the questions as below:
>
> ---
> # Weakness
> ### 1. Regarding the Notations
> - We will revise the manuscript to better describe the notations used. $Y_U$ is the set of labels corresponding to unlabeled data; $X^*$ and $Y^*$ are the decision variables representing candidate data (image) and corresponding labels to be added to the labeled set. $Q_0$ is the initial query size.
>
> ### 2. Regarding the Presentation Clarity
> - We thank the reviewer for this suggestion. We will gladly elaborate on lines 99-100 in the next revision. For now, we elaborate here. It is true that both SL and SSL suffer when classes are imbalanced; performance degradation depends on the ratio between classes. What we intended to describe in lines 96-98 is that even with balanced classes, SL fails to achieve meaningful performance while SSL can. When classes are imbalanced which inevitably occurs in our more practical AL (ASSL) setting, SL still fails while only recently have SSL algorithms that appropriately incorporate class imbalance been devised.
>
> ### 3. Regarding the Clarity on Algorithm
> - We will expand on lines 188-195 to better describe how NPC’s using directions in addition to magnitude affects its choice of candidate labels.
>
> ### 4. Regarding the Originality
> - We would like to clarify our original contributions. Our work establishes two goals: to propose a new experimental method to measure the label-complexity of active learning algorithms instead of data (image) and label-complexity; to design a new active learning algorithm that reduces the number of labels required to attain high performance. The use of semi-supervised learning is not meant to improve the performance of active learning regarding our above goal and is rather proposed to evaluate the label-complexity of any active learning algorithms, not just ours. We do not consider this to be our algorithmic contribution. In our view, our algorithmic contribution warrants novelty because our work extends the usage of gradient embedding vectors, which had been motivated by its capturing of uncertainty information, to consider all three aspects in Sec. 3.2 based on our observation that the three motivations can all be addressed by relating to the network’s training dynamics. While these three have been called for in previous literature and addressed independently in respective problems of interest, we unified the three into a common concept (NPC) whose efficacy to active learning was demonstrated.
> ---
>
> # Questions
> ### 1. Regarding the Fig.1
> - Fig. 1 is our batch extension to the ranking comparisons in (Fig. 2, Huang et al., 2016) and (Fig. 3, D. Gissin and S. Shalev-Schwartz, 2019) used to compare how similar algorithms are in selecting candidate data. A pair of algorithms with high IoU implies that the two algorithms behave similarly, whereas a pair with low IoU implies drastic methodological differences of the two.
>
> ### 2. Regarding the Implementation
> - Yes, as the reviewer carefully noted, this part describes the preprocess of selecting the labeled training data. We will make that more clear in the revision.
>
> ### 3. Regarding the SSL as the Better Benchmark for Measuring Label-complexity
> - As described in the first part of the introduction, the problem of interest is pool-based AL where both unlabeled and $Z_U = X_U$ and labeled data $Z_L = (X_L, Y_L)$ are available and there is a restriction on how many labels can be acquired. The goal of AL in this setting is to maximize performance using the fewest possible number of labels. Our sentence quoted by the reviewer emphasizes how previous experimental designs in AL research had instead been evaluating the performance gains from adding both data (images) and labels, instead of only the latter as the reviewer notes. We are interested in measuring performance gains from only newly acquired labels, and therefore we described why SSL is more appropriate in measuring label-complexity, i.e. performance gains from only newly acquired labels.

---

### Official Review · Reviewer_1Ma5 · 2022-07-17

**Rating:** 3
**Confidence:** 3
**Soundness:** 3 good
**Presentation:** 1 poor
**Contribution:** 2 fair

**Summary:**

This paper focuses on the active learning (AL) paradigm to reduce the label complexity compared to typical deep learning methods and proposes to capitalize on the semi-supervised learning (SSL) context instead of the typical supervised learning (SL) one, to be combined with AL. To that end, the authors advocate a neural pre-conditioning (NPC) algorithm  that relies on neural tangent kernels (NTKs) leveraging the minimum eigenvalue of the Gram matrix induced by NTKs. Empirical results demonstrate that merging NPC with some well-motivated SSL approaches can lead to improved prediction performance utilizing only few labeled data.


**Questions:**

- In line 25 in the sentence “Given images X and a labeling budget B, what’s the maximum performance that can be achieved?”. Where does the performance refer to? Does it refer to a classification task (such as object categorization) or regression (or both)?
- In lines 29-33, the sentence “SL fails to extricate…newly-acquired labels“ is not clear. Does it just mean that SL does not leverage $\mathcal{Z}_U$ during training?
-  In lines 55-56, in the sentence “Because of the difference..”, where does the “difference” refer to?
-  In line 129, in the sentence “In AL, it is impossible to know acquire labels while ensuring” what is the main verb (know or acquire)? Is it maybe to know acquired labels?
-  In lines  156-158, the sentence “We show after presenting our algorithm in Sec. 3.3 that our selection criterion does indeed select based on the gradient’s direction which captures uncertainty information, which…” is incomplete and unclear. What is selected?
-  I believe that in Eq. 1 there is a typo. Instead of $\mathcal{Y}^* \subset \mathcal{X}_U$ is it maybe $\mathcal{Y}^* \subset \mathcal{Y}_U$?
-  In lines 34-35, it is mentioned that “Performance gains as a SL algorithm incrementally acquires labels remain minor relative to passive learning (PL).” This is not true in general. There exist AL-SL based  methods with well-documented merits compared to passive approaches; see e.g [1], [2].
-  In line 35, it is mentioned that “This may be attributed to the large numbers of labels required to attain meaningful  accuracy for SL…”. However, this does not always hold; see e.g [1], [2].
-  In lines 38-39, it is mentioned that “While in principle more data should always be better…”. However, this does not always hold especially when the data are noisy (or redundant).

Some examples of more minor issues are listed below

- In line 5, instead of “Although unlabeled data is readily…”, it should be “Although unlabeled data are readily…”.
- In line 27, instead of “have been evaluated by measuring downstream supervised…” it should be “have been evaluated by measuring the downstream supervised”.
- In line 28, instead of “but we argue downstream semi-supervised learning (SSL) performance...” it should be “but we argue that the downstream semi-supervised learning (SSL) performance...”.
-  In line 69, the verb “have” appears twice.
-  In line 114, instead of “computes” it should be “compute”.

 I strongly suggest that the authors go over the paper carefully and correct the aforementioned and other similar grammatical/syntax errors.

[1] Kapoor, Ashish, Kristen Grauman, Raquel Urtasun, and Trevor Darrell. "Active learning with gaussian processes for object categorization." In 2007 IEEE 11th international conference on computer vision, pp. 1-8. IEEE, 2007.

[2] Pasolli, Edoardo, and Farid Melgani. "Gaussian process regression within an active learning scheme." In 2011 IEEE International Geoscience and Remote Sensing Symposium, pp. 3574-3577. IEEE, 2011.


**Limitations:**

The presentation of the limitations is good in general. Some additional concerns of mine are listed above.

**Strengths And Weaknesses:**

STRENGTHS

- The proposed framework seems to work well in practice.
- The literature review is good.

WEAKNESSES

- The writing and presentation of the paper admit further improvements. There are many parts of the paper that are hard to follow (see the Questions section below for some examples).
- The novelty of the paper is somewhat limited. The SSL methods used are well known and the use of the Gram matrix (and its properties) in NTKs has been explored in the literature. In addition, besides the empirical performance, the contributions of the advocated approach need to be highlighted more clearly.

---

> ### Author Response · Authors · 2022-08-01
> **Author Response to Reviewer 1Ma5**
>
> Thank you very much for your helpful comments for improving the presentation of our work.
>
> We hope our correction of grammatical errors and clarification on our work’s novelty addresses the reviewer’s concern, and we look forward to discussing any further questions.
>
> ---
> # Weakness
> ### Regarding Writing and Presentation
> - Thank you for the detailed suggestions on wording. We will incorporate them in our next revision as answered below.
>
> ### Regarding the Novelty
> - Our contribution is two-fold. One is to use semi-supervised learning classification performance to measure the efficacy of active learning algorithms, i.e. how much they reduce *label complexity*. We agree that there is no technical novelty involved, but we see this as an important distinction between measuring generic sample complexity vs. label complexity. We view our algorithmic design, however, to be a non-trivial extension to existing active learning algorithms. Existing work has sought to establish a proxy to a model’s uncertainty (Sec. 3.2.1), an active learning algorithm that operates in the batch-mode, and construction of datasets to maximize generalization (classification) performance. These have been studied extensively in distinct fields, but we identify how the NTK can commonly address all these motivations simultaneously. Furthermore, the usage of NTK was originally established to study gradient descent dynamics and did not come with immediate applications to active learning, whose connection is, to our view, made in our work.
>
> ---
>
> # Questions
> ### Regarding the Sentence in Line 25
> - Our work considers classification accuracy.
>
> ### Regarding the Sentence in Lines 29-33
> - This sentence does refer to SL’s not using $X_U$, but emphasizes its usage as an evaluation metric for active learning performance.
>
> ### Regarding the Sentence in Lines 55-56
> - The difference refers to the difference between methods by ours vs. Huang et al. (2016) and Ash et al. (2020). Specifically, they use the gradient’s norm and gradient vectors + k-means++; we build a Gram matrix by taking pair-wise inner products of gradient vectors as its entries.
>
> ### Regarding the Sentence in Line 129
> - The typo will be concisely corrected to “it is impossible to ensure balanced classes in either…”.
>
> ### Regarding the Sentence in Lines 156-158
> - This will be revised to say “select *data to be labeled* based on the gradient’s direction”.
>
> ### Regarding the Equation (1)
> - Eq. (1) is minimizing over subsets of the cartesian product between $X_U$ and $Y_U$. It was originally written as if they are a tuple, but will be corrected to avoid confusion.
>
> ### Regarding the Sentence in Lines 34-35
> - We thank the reviewer for pointing this out.  We were only referring to deep learning-based AL as suggested in the first sentence in abstract; we will make this clear in our revision.
>
> ### Regarding Sentence in Lines 38-39
> - Ideally in principle, more data cannot reduce information even if they are noisy. Whether in practice noisy data can help is a different question; we will fix the wording as we do agree our wording can sound confusing.
>
> ### Regarding Minor Issues on Grammatical Errors
> - We again thank the reviewer for pointing out typos and ambiguous wordings.

---

> > ### Comment · Reviewer_1Ma5 · 2022-08-09
> > **Response to the authors**
> >
> > I would like to thank the authors for their response and I appreciate their effort for addressing my comments. However, the presentation and the writing admit further improvements (The points mentioned were indicative ones). In my opinion, the paper in its current form does not merit publication.

---

### Comment · Area_Chair_9odQ · 2022-08-10
**Reminder**

Dear reviewers,

Please go through the rebuttal (if you have not) and acknowledge that you have done so. Thanks!

AC

---

### Meta-Review · Area_Chair_oeC4 · 2022-08-31

**Recommendation:** Accept
**Confidence:** Certain

**Metareview:**

This paper proposes an NTK-based active learning method that, when combined with SSL methods, performs well on a number of tasks.  There are several passages in the paper that reviewers feel need to be improved for clarity.  The authors respond to these issues, and I think the responses can be transferred to the camera ready.


**Award:**

No

---

### Decision · Program_Chairs · 2022-09-14

Accept